# COVID-19 Vaccine Uptake and Hesitancy among Pregnant and Lactating Women in Saudi Arabia

**DOI:** 10.3390/vaccines11020361

**Published:** 2023-02-05

**Authors:** Hayfa A. AlHefdhi, Syed Esam Mahmood, Manar Ahmed I. Alsaeedi, Haifa’ Hisham A. Alwabel, Mariam Salem Alshahrani, Ebtihaj Yahya Alshehri, Rawan Ahmed O. Alhamlan, Maram Nawar Alosaimi

**Affiliations:** 1Family and Community Medicine Department, College of Medicine, King Khalid University, Abha 62529, Saudi Arabia; 2Intern, College of Medicine, King Khalid University, Abha 62529, Saudi Arabia

**Keywords:** COVID-19 vaccine, pregnancy, breastfeeding, attitudes, behavior, Saudi Arabia

## Abstract

Introduction: Pregnant and breastfeeding women comprise a high-risk group for the development of severe COVID-19. Therefore, vaccination is highly recommended for perinatal women; however, vaccination levels for this group remain inadequate. This study explores the percentage of COVID-19 vaccination among Saudi pregnant and lactating women, as well as their attitudes toward it. Methods: We conducted a cross-sectional questionnaire-based survey on a sample of Saudi pregnant and breastfeeding women. The study included pregnant and lactating women. Results: The percentage of COVID-19 vaccine uptake was 78.2%. A total of 45 (21.8%) out of 206 women did not receive the vaccine. The overall vaccine hesitancy was 21.8%. Breastfeeding women were 2.86 more likely not to receive the vaccine as compared to pregnant women. Being a mother of over five children increased the vaccine uptake among our participating women (*n* = 20, 90%; *p* < 0.01). The majority of the subjects had taken the Pfizer vaccine (81.98%, 132/161). The availability of the COVID-19 vaccine was the most common factor for choosing a particular vaccine. Protection from infection (60.2%, 97/161) was reported as the main driver for vaccine uptake. The most common reason perceived for delaying COVID-19 vaccination was being worried about the side effects (176, 85.44%) on one’s own body and the effects on the unborn child (130, 63.1%). Conclusion: We uncovered high levels of hesitancy, primarily induced by concerns about adverse effects and social media-related misinformation. These high levels of vaccine uptake are likely due to the large-scale obligatory vaccination program provided in Saudi Arabia, which was well-structured and far reaching. Our results provide further support for the so-called “protection motivation theory” in boosting vaccine acceptance. Counseling and educating pregnant and breastfeeding women about COVID-19 vaccination is the need of the hour.

## 1. Introduction

Coronavirus disease 2019 (COVID-19) is an infectious disease caused by the SARS-CoV-2 virus. The most recent global data indicate that nearly 600 million people were infected, and 6.5 million deaths were registered between 2020 to 2022 [1]. Most people infected with this virus encounter mild-to-moderate illness, and recovery is likely not to require any special treatment. People at risk of severe disease are those aged 65 years and older and patients with comorbid medical conditions [2]. Pregnant women are a high-risk group for the development of acute respiratory syndrome [3]. Notably, an increased risk for caesarean section, preterm birth, and neonatal intensive care unit admissions in pregnant women that tested positive for SARS-CoV-2 has been observed [4]. Therefore, vaccination is highly recommended for perinatal women [5]. However, vaccination among pregnant women is inadequate [6]. Substantial hesitancy has been reported among the general population worldwide and in Saudi Arabia [7,8]. The WHO definition of “vaccine hesitancy” considers hesitancy if the time of vaccination has been delayed, or not vaccinated [9]. A recent study conducted in Saudi Arabia reports that about 12.8% of children have not received the vaccination, 55% of parents have some sort of hesitation and 32.2% of parents did not hesitate before vaccinating their children against COVID-19 [10]. Vaccine newness, as a reason for hesitating to get vaccinated, was most reported among non-vaccine supporters in another study from Saudi Arabia [11]. The side effects of the COVID-19 vaccine are the most important barrier to vaccine acceptance [12]. The rates of vaccine hesitancy in pregnant and breastfeeding women across high-income countries or regions ranged from 7% to 77.9%, with an average of 48% [13].

Researchers have acknowledged that pregnant and lactating women receive mixed messages in terms of safety of COVID-19 vaccination, primarily due to their exclusion from many trials [14], thus contributing to hesitancy and reluctance among perinatal women in terms of vaccination. The studies on this particular issue are scarce and, to the best of our knowledge, no study has been reported from the Aseer region to date, especially taking into account the pregnant and breastfeeding females. Therefore, we have attempted to assess the COVID-19 vaccine hesitancy among pregnant and lactating women attending the antenatal care and outpatient clinics at the Maternity and Children Hospital in Abha, Saudi Arabia.

The main objective of the current study was to assess levels of hesitancy, attitudes, and practices of women in the pregnancy and breast feeding period regarding COVID-19 vaccination.

## 2. Methodology

Study setting and design: Abha is one of the beautiful cities in the province of the Aseer region of Saudi Arabia, which is situated on the slopes of the Sarawat mountains. Abha has a population of about 366,551. This cross-sectional study was conducted among pregnant and breastfeeding women attending the antenatal care and outpatient clinics at the Maternity and Children Hospital, Abha, Aseer Region, KSA.

Study duration: The duration of the study was nine months (April–December 2021), inclusive of four weeks for preparing the study tool, three months for data collection, and one month for data analysis.

Study population: The study population consisted of all female pregnant and breastfeeding women attending the antenatal care and outpatient clinics at the Maternity and Children Hospital in Abha. We included Saudi pregnant women (last trimester) and breastfeeding women (with infants aged less than 6 months), aged between 18 and 49 years, who attended the antenatal care and outpatient clinics at the hospital between April to December 2021, and agreed to participate in the study. We excluded non-Saudi women, those aged younger than 18 or older than 49, and women not in the perinatal period (i.e., neither pregnant nor lactating mothers). About 30 pregnant females attend the outdoor clinics of the hospital each day. As data collection was done for about 3 months, we assume that around 2700 patients attended the clinics. However, our study included only pregnant females in the last trimester and the lactating females having infants aged less than 6 months. We included 206 females in our survey.

Sampling technique: A convenience technique was used to select 10–15 study participants from different antenatal care and outpatient clinics. The women were only included once in the study, and they had the right to refuse participation.

Data collection: A structured questionnaire was used to collect data from the pregnant and breast feeding women exiting the antenatal care and outpatient clinics at the hospital. Data was collected using an adapted and modified questionnaire from the existing studies (8, 19). We set out in this study to develop and validate an instrument to use for the assessment for vaccine hesitancy in pregnant women. A pilot survey was conducted on 30 pregnant females before initiating the actual data collection; however, these pilot samples were excluded from the final sample size. The prime objective of the pilot survey was to guarantee the validity and reliability of the questionnaire. The face and content validity of the questionnaire was assessed by the principal and co-investigators themselves. Face validity was evaluated through the review and comments offered by a panel of experts related to readability, clarity of wording, layout, and feasibility of the questionnaire. Content validity was evaluated by the content validity index, which is the mean content validity ratio of all questions in a questionnaire. The paper-based questionnaire consisted of 22 items. The questionnaire was translated from English to Arabic (local language) by a bilingual person to enable an easy understanding of the questions and avoid any questionnaire bias.

Operational definitions: The WHO definition of “vaccine hesitancy” considers hesitancy if the time of vaccination has been delayed or not vaccinated. Vaccinated means fully vaccinated, i.e., who has received two doses of COVID-19.

Data analysis: The dataset was arranged into paper-based questionnaire packs and then entered into an excel spreadsheet by the principal investigator. The data were organized into successive columns, and R software (version 4.1.2) was used for comprehensive statistical analysis. Exploratory analysis was achieved using proportions and mean and standard deviation measures. The associations have been exposed through odds ratios. *p*-values less than 0.05 are considered statistically significant. Tables were constructed based on the exploratory statistical results. Pie and bar charts were used to visually display the results. Multinomial logistic regression was performed to estimate the parameters.

Ethical permission to conduct this research was obtained from the local Institutional Review Board Committee of King Khalid University.

## 3. Results

This study included 206 pregnant and breastfeeding women. The demographic results are shown in Table 1, in addition to the unadjusted effect on receipt of the COVID-19 vaccine. Notably, none of the variates of age, education, or employment exerted a significant effect on the uptake of the COVID-19 vaccine.

The majority of the study subjects belonged to the 20–29- and 30–39-year age groups. Most of the study subjects had a bachelor’s degree (67.4%, 139/206), of which 78.4% had received the COVID-19 vaccine. Of the total, 80.4% of those employed and 77.5% of those unemployed had received the COVID-19 vaccine.

Table 2 shows the association between the medical comorbidities of the study participants and their COVID-19 vaccination status. No statistically significant association was found for any effect regarding medical conditions and uptake of the COVID-19 vaccine. In total, 32/206 (15.5%) of the study subjects had a chronic disease, whereas 12/206 (5.8%) had a mental illness. The most common chronic diseases were asthma (11/32, 34.3%) and hypothyroidism (10/32, 31.25%). Among the mentally ill study subjects, the majority suffered with depression (6/12, 50%).

Table 3 shows the characteristics of those women who did and did not receive the COVID-19 vaccine during pregnancy. The uptake of the COVID-19 vaccine was far better among pregnant women (*n* = 118, 84.3%) than among breastfeeding women (*n* = 43, 65.2%; *p* < 0.01). Breastfeeding women were 2.86 times more likely not to receive the vaccine as compared to pregnant women, OR = 2.86 (95%CI, 1.45–5.66). Moreover, a considerable proportion of women with over five children received the vaccine compared to women with under five children (*n* = 20, 90%; *p* < 0.01). Of the total, 51/206 (24.7%) study subjects reported complications during their last pregnancy. Bleeding was the most common (*n* = 12/51, 23.5%). Seventy-five subjects were diagnosed with COVID-19 during or after pregnancy and 63/75 (84%) had received the COVID-19 vaccine, of which only 6/75 (8%) were admitted.

Figure 1 shows that the 161 (78.2%) and 45 (21.8%) subjects did or did not receive the COVID-19 vaccine, respectively. Out of those unvaccinated, 20%, (9/45) wanted to receive the vaccine as soon as possible. A higher proportion of the subjects (35.5%, 16/45) were willing to get vaccinated after delivery. Additionally, a moderate proportion (3/45, 6.6%) were willing to get vaccinated before traveling, and nearly 26.6% (12/45) had decided not to receive the COVID-19 vaccine. The WHO considers hesitancy if the time of vaccination has been delayed or not vaccinated. Therefore, the vaccine hesitancy is = 21.8%.

Table 4 shows the reasons for choosing the particular COVID-19 vaccine. Most of the subjects had received the Pfizer vaccine (81.98%, 132/161), followed by the AstraZeneca Oxford (14.2%, 23/161) COVID-19 vaccine. Very few had received the Moderna COVID-19 vaccine (3.1%, 5/161). The availability of the COVID-19 vaccine was the most common factor for choosing a particular vaccine, followed by the factors of “heard through social media” and “having less complications”.

Figure 2 shows the reasons for receiving the COVID-19 vaccine. Protection from infection (60.2%, 97/161) was the foremost reason influencing the subjects. Of the total, 25.4% (41/161) of the subjects had received the COVID-19 vaccine to enter public places. Going back to work (13.6%, 22/161) also encouraged the subjects to receive the COVID-19 vaccine.

Table 5 shows the perceptions of the study subjects toward the COVID-19 vaccine. Overall, 192 out of 206 (93.2%) females had delayed their vaccination due to uncertainty of the safety of the vaccine and some other medical reason at some point of time. The most common reason for delaying receiving the COVID-19 vaccine was worry about the side effects (176, 85.44%). Almost 45% of the subjects believed that the COVID-19 vaccine was effective. Half of the subjects felt comfortable while receiving the COVID-19 vaccine. Social media (125, 60.68%) was the most common source of information for the vaccine, followed by a physician’s advice (54, 26.21%). The majority of the subjects (130, 63.11%) were fearful about receiving the COVID-19 vaccine in case it affected the fetus.

Table 6 shows the different perceptions of the study subjects according to the COVID-19 vaccines. In the present study, the vaccines of different pharmaceutical companies taken were AstraZeneca Oxford, Moderna, Pfizer and those who had not taken any vaccine were excluded. The Pfizer COVID-19 vaccine was quite popular among the participants, with 72 (78.3%) believing in its effectiveness and a further 77 (77.8%) being comfortable receiving it. There was no difference in terms of the source of information about the different vaccines.

Table 7 shows the multinomial logistic regression to estimate the effect of various influencers over the selection of different COVID-19 vaccines. Comfortability was the significant cause for vaccine selection in the case of AstraZeneca Oxford and Moderna, while in the case of Pfizer, availability was the significant reason (*p* = 0.005). No other cause was found to be significant for a particular vaccine selection.

## 4. Discussion of the Key Findings

The current study surveyed a large sample of 206 pregnant and breastfeeding Saudi women. We found a 74.8% prevalence of COVID-19 vaccination among them. This equates to three out of every four perinatal women in Saudi Arabia. A lower acceptance level (68%) of the COVID-19 vaccine was reported among pregnant women in Saudi Arabia in 2021 [15]. In another study by Samannodi et al., more than half of the study sample (57.1%) had received the COVID-19 vaccination among pregnant women and those who are planning for pregnancy in Saudi Arabia [16]. This difference can be attributed to the fact that our study data is recent, and the vaccine acceptance might have improved. The Saudi government and ministry of health have been encouraging pregnant women to be vaccinated, as recommended by the WHO [16]. The acceptance level in our study is well above the prevalence of 13.4% for COVID-19 vaccination found recently among pregnant women in Japan [17]. However, the difference may be attributable to the different social classes and levels of education in these studies. Similarly, the current vaccine hesitancy level among the women we investigated was 21.8%, similar to the vaccine hesitancy level (32%) reported by a previous Saudi study [15].

Clearly, the Kingdom provided a large-scale vaccination program that was well-structured and far reaching. Moreover, vaccination was a pre-requisite for visiting public spaces and performing Omrah. Such measures were reflected in the high level of awareness and, thus, the uptake of COVID-19 vaccination in Saudi Arabia, even in the early stages of the pandemic [16]. The country-specific portrayal of the importance of the vaccine has been shown to boost the acceptance rates among pregnant women [17]. We noted that background education and employment did not substantially impact vaccine uptake, nor did medical and psychiatric comorbidities. This may point toward a higher influence for governmental measures on vaccine acceptance among Saudi women, far more than individual differences. Some papers have found a link between good education and better knowledge about the pros and cons of vaccination and, therefore, better vaccine acceptance among perinatal women [15,18]. More efforts to improve confidence and trust in the vaccine among women with higher educational achievement are required to improve vaccine uptake [19].

One striking finding we uncovered was a correlation between the number of children and the mother’s acceptance of COVID-19 vaccination. A similar finding was reported in a survey of mothers of young children in Poland [20]. That was regarded as a positive association, as the acceptance of the COVID-19 vaccination by mothers is expected to influence positive attitudes toward vaccination in children as well, with clear intentions to facilitate their vaccination [21].

The prevalence of COVID-19 infection among our sample was 36.4%, as only 75 women reported it, with varying degrees of severity. These is well above the crude prevalence of 2.5% reported among pregnant women in the USA [22], but close enough to the 25% figure reported in a recent Mexican survey [23]. Our figure is presumed to be inflated with COVID-19 infection cases that happened prior to pregnancy or during lactation. A report from Madinah in Saudi Arabia found that 50.2% of pregnant women were seropositive for COVID-19 [24].

The vaccine hesitancy level was 21.8% among our participating pregnant women. In an Iranian study, some 42.6% of pregnant women were hesitant to receive the COVID-19 vaccine, mostly because of personal gynecological issues and skepticism with regard to its benefits [25]. The major reason reported by a Saudi study for refusing the COVID-19 vaccination was a lack of data about COVID-19 vaccination safety (76%) [15].

Social media, perception of fewer adverse effects, and availability were the main drivers for women seeking a particular COVID-19 brand in this study. Clearly, social media has exerted a profound effect on the public perception of COVID-19 vaccination, an observation that resulted in numerous researchers calling for healthcare workers to utilize social media in dispelling misinformation [26]. Social media can be utilized effectively to encourage vaccine uptake, and research should evaluate the effectiveness of well-designed social media campaigns on vaccine hesitancy and acceptance. Indeed, misinformation that inflates the likelihood of vaccine side effects can severely hamper vaccine acceptance among the public [27]. Recent surveys, such as our current investigation, have consistently confirmed a long-lasting effect of protection dynamics on vaccine uptake behavior [28].

The primary facilitator of vaccine acceptance among the participants was belief in protection from COVID-19. Our results provide further support for the so-called “protection motivation theory” in boosting vaccine acceptance [29]. This is a long-proposed theory that postulates that belief in the seriousness of the “individual threat appraisal” of COVID-19 and in the potential protection of the vaccine and uptake of the vaccines are the “occurrence of desired behaviors” [30].

Preoccupation with side effects was the main driver of COVID-19 vaccine hesitancy among our subjects. This confirms the findings of pan-continental surveys that mistrust in healthcare systems and concerns about serious adverse effects are the two main barriers against the uptake of the COVID-19 vaccine [31]. Other studies have suggested that cost can be a deterrent, particularly among pharmacists and policymakers [32]. Transparency in terms of reporting rare side effects, in addition to the development of modified vaccines, have been suggested as options to overcome public fears of vaccination [33]. Reports have indicated that adverse effects remain extremely rare and incomparable to the benefits of vaccines [34].

The majority (63.1%) of the women were particularly apprehensive of the vaccine’s effect on the developing fetus. This finding is indeed not an exception in the literature. Over 85% of Japanese women expressed concern about the potential negative effects of the COVID-19 vaccination on fetal well-being [17]. Similarly, 51.9% of Saudi pregnant women refused to take the vaccine because of the possibility of harm to their baby [15]. Scientifically speaking, vaccines are reported to be quite safe among pregnant and lactating women [35,36]. Furthermore, anti-COVID-19 virus antibodies have been noted to be developed by infants following maternal vaccine uptake [37]. Such encouraging information should be made available to the public through official media channels and healthcare-related social media outlets. It is crucial to provide information in simple Arabic on the safety of COVID-19 vaccines for children and pregnant women so that COVID-19 vaccine administration can continue efficiently. Such accessible information may also help to minimize hesitancy regarding vaccinations and increase vaccination uptake [38].

We also found that the Moderna vaccine was the least popular among our sampled participants. Many believed it is not effective and a further considerable proportion were unsure about its effectiveness. This is not unique to our participants. It has been noted that hesitancy was vaccine-specific among French citizens [39]. It is difficult to explain such disparities in terms of vaccine acceptance. Moderna is an mRNA-1273 COVID-19 vaccine with a high level of protection against severe COVID-19 disease and hospitalization [40]. However, it is likely that official media and social media labeling of Moderna was indicative of an erroneous suboptimum effect compared to other vaccines. A recent systematic review and meta-analysis assessed the effectiveness and safety of coronavirus disease 2019 (COVID-19) vaccines (including the BNT162b2 vaccine, mRNA-1273 vaccine, and adenovirus vector vaccine) for pregnant women in real-world studies. They found that messenger-RNA vaccines could reduce the risk of infection in pregnant women (OR = 0.13, 95% CI, 0.03–0.57). No adverse events of COVID-19 vaccination were found on pregnant, fetal, or neonatal outcomes [41]. It has been recently reported that vaccination during pregnancy builds antibodies that can help protect the baby; however, more data is needed in support [42,43]. Factors reported for increased vaccine hesitancy among pregnant females are lack of trust and hearing or reading about negative events from different sources [44]. Correcting the misinformation and replacing it with the accurate one can diminish the continued influence of misinformation amongst such females. To the best of our knowledge, this is one of the first studies to assess vaccine hesitancy among pregnant women in the region. Evidence gaps still remain around COVID-19 vaccines in pregnancy, highlighting the need for further investigation. Counseling and educating pregnant and breastfeeding women about COVID-19 vaccination is the need of the hour. This shall help the government and policymakers to prevent unwanted pregnancy and birth outcomes and improve the overall maternal and child health.

## 5. Strengths and Limitations of the Study

The large sample size, including both pregnant and lactating females being extensively analysed, and use of a validated research tool are the two main strengths of the current study. The few significant limitations include desirability bias, non-random sampling, and cross-sectional design. In addition, this study was conducted at one center in Saudi Arabia and our findings may not present the vaccine hesitancy across Saudi Arabia. We hope in the future to have all the required resources to do multicentric/nationwide studies.

Future research needs to be qualitative in nature and preferably use longitudinal measuring of attitudes at several time points to robustly evaluate the barriers and facilitators of COVID-19 vaccination.

## 6. Conclusions

Although the COVID-19 vaccination uptake among Saudi women was in line with the global rates (three out of four), we uncovered high levels of hesitancy (91.3%), primarily induced by concerns about adverse effects and social media-related misinformation. The high level of vaccine uptake was likely due to the large-scale obligatory vaccination program provided in Saudi Arabia, which was well structured and far-reaching. The primary facilitator of vaccine acceptance among the participants was belief in protection from COVID-19. Our results provide further support for the so-called “protection motivation theory” in boosting vaccine acceptance.

## Figures and Tables

**Figure 1 vaccines-11-00361-f001:**
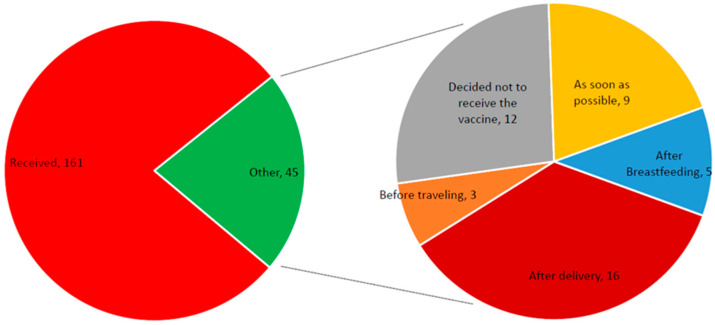
Number of subjects who did and did not receive the COVID-19 vaccine.

**Figure 2 vaccines-11-00361-f002:**
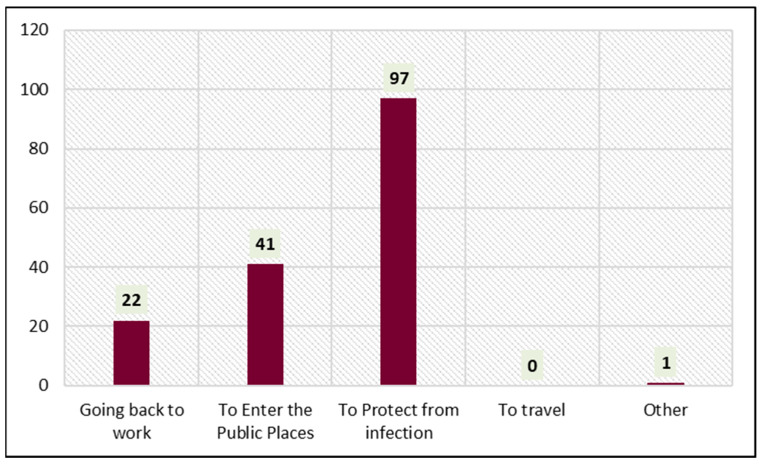
Reason for receiving the vaccine.

**Table 1 vaccines-11-00361-t001:** Sociodemographic profiles of the COVID-19 vaccine recipients.

Variables		COVID-19 Vaccine Received		Total	*p*-Value
		No (*n* = 45)	Yes (*n* = 161)		
Age (in years)	<20	0	3	3	0.59
		0.0%	100.0%	100.0%	
	20–29	20	80	100	
		20.0%	80.0%	100.0%	
	30–39	22	64	86	
		25.6%	74.4%	100.0%	
	40–49	3	14	17	
		17.6%	82.4%	100.0%	
Education	High school	13	49	62	0.60
		21.0%	79.0%	100.0%	
	Bachelors	30	109	139	
		21.6%	78.4%	100.0%	
	Masters	2	3	5	
		40.0%	60.0%	100.0%	
Employed	No	36	124	160	0.67
		22.5%	77.5%	100.0%	
	Yes	9	37	46	
		19.6%	80.4%	100.0%	
Type of employment	Government	7	26	33	0.56
		21.2%	78.8%	100.0%	
	Unemployed	36	124	160	
		22.5%	77.5%	100.0%	
	Private	2	11	13	
		15.4%	84.6%	100.0%	

**Table 2 vaccines-11-00361-t002:** Association between COVID-19 vaccination status and comorbidities of the study subjects.

Comorbidities		COVID-19 Vaccine Received		Total	*p*-Value
		No (*n* = 45)	Yes (*n* = 161)		
Chronic disease	No	37	137	174	0.63
		21.3%	78.7%		
	Yes	8	24	32	
		25.0%	75.0%		
Type of chronic disease	Anemia	0	1	1	0.44
		0.0%	100.0%		
	Asthma	3	8	11	
		27.3%	72.7%		
	Diabetes	0	7	7	
		0.0%	100.0%		
	Hypercoagulation	0	1	1	
		0.0%	100.0%		
	Hypothyroidism	4	6	10	
		28.6%	71.4%		
	Kidney disease	1	0	1	
		100.0%	0.0%		
	PCOD	0	1	1	
		0.0%	100.0%		
	No	37	137	174	
		21.3%	78.7%		
Mental illness	No	42	152	194	0.78
		21.6%	78.4%		
	Yes	3	9	12	
		25.0%	75.0%		
Type of mental illness	Anxiety	1	2	3	0.09
		33.3%	66.7%		
	Depression	0	6	6	
		0.0%	100.0%		
	Depression and panic	1	0	1	
		100.0%	0.0%		
	Panic	0	1	1	
		0.0%	100.0%		
	Schizophrenia	1	0	1	
		100.0%	0.0%		
	No	42	152	194	
		21.6%	78.4%		

**Table 3 vaccines-11-00361-t003:** Characteristics of women who did and did not receive COVID-19 vaccine during pregnancy.

Variables		COVID-19 Vaccine Received		Total	*p*-Value
		No (*n* = 45)	Yes (*n* = 161)		
Pregnancy status	Breastfeeding	23	43	66	<0.01
		34.8%	65.2%	100.0%	
	Pregnant	22	118	140	
		15.7%	84.3%	100.0%	
Ever diagnosed with COVID-19 during or after pregnancy	No	33	98	131	0.21
		25.2%	74.8%	100.0%	
	Yes	12	63	75	
		16.0%	84.0%	100.0%	
If yes, admitted or experienced any complications	No	10	59	69	0.68
		14.5%	85.5%	100.0%	
	Yes	2	4	6	
		33.3%	66.7%	100.0%	
	Not diagnosed with COVID-19	33	98	131	
		25.2%	74.8%	100.0%	
Total number of pregnancies	1	8	65	73	<0.01
		11.0%	89.0%	100.0%	
	2–5	35	76	111	
		31.5%	68.5%	100.0%	
	>5	2	20	22	
		9.1%	90.9%	100.0%	
Abortion or still birth	No	31	109	140	0.88
		22.1%	77.9%	100.0%	
	Yes	14	52	66	
		21.2%	78.8%	100.0%	
Complications during last pregnancy	Eclampsia	0.0	2	2	0.83
		0.0%	100.0%	100.0%	
	Hypertension	1	4	5	
		20.0%	80.0%	100.0%	
	Bleeding	2	10	12	
		16.7%	83.3%	100.0%	
	Diabetes	1	11	12	
		8.3%	91.7%	100.0%	
	Other	6	14	20	
		30.0%	70.0%	100.0%	
	No	35	120	155	
		22.6%	77.4%	100.0%	

**Table 4 vaccines-11-00361-t004:** Reason for choosing the particular COVID-19 vaccine.

Vaccine	As per Availability	Having Less Complications	Heard Through Social Media	Not Taken	Recommended by Friends	Recommended by Doctor	Total
AstraZeneca Oxford	17	0	1	0	2	3	23
Moderna	4	0	0	0	0	1	5
Not taken	0	0	0	45	0	0	45
Outside the kingdom	1	0	0	0	0	0	1
Pfizer	72	22	25	0	8	5	132
Total	97	22	26	45	10	9	206

**Table 5 vaccines-11-00361-t005:** Perceptions of the COVID-19 vaccine.

Perceptions toward the COVID-19 Vaccine	Factors	*N*	%
Reason for delaying receiving the vaccine	Family refusal	1	0.49
	Medical advice	3	1.46
	Want to take particular type and avoid other types	7	3.40
	Allergy	1	0.49
	Husband’s refusal	1	0.49
	Medical contraindication	3	1.46
	Not due	6	2.92
	Vaccine is not available	8	3.88
	Worried about the side effects	176	85.44
COVID-19 vaccine effective	No	45	21.84
	Yes	92	44.66
	Not sure	69	33.50
Comfortable receiving the vaccine	No	53	25.73
	Yes	99	48.06
	Not sure	54	26.21
Source information for the vaccine	Friends	5	2.43
	Close family members	10	4.85
	Physician	54	26.21
	Distant relatives	12	5.83
	Social media	125	60.68
Fear of receiving the vaccine	COVID-19 is not considered a dangerous disease	1	0.49
	Vaccine will affect the fetus	130	63.11
	COVID-19 vaccination can cause infection	2	0.98
	Family hesitancy	6	2.91
	Not enough studies on pregnant women	44	21.36
	Vaccine effectiveness is low	5	2.43

**Table 6 vaccines-11-00361-t006:** Association between the perceptions and COVID-19 vaccines.

Perceptions toward the COVID-19 Vaccine	Factors	AstraZeneca Oxford	Moderna	Pfizer	*p*-Value
		(*n* = 23)	(*n* = 5)	(*n* = 132)	
Reason for delaying receiving the vaccine	Medical advice	0	0	2	0.239
		0.00%	0.00%	1.52%	
	Not due	2	0	4	
		8.70%	0.00%	3.03%	
	Want to take particular type and avoid other types	1	0	4	
		4.35%	0.00%	3.03%	
	Allergy	0	0	1	
		0.00%	0.00%	0.76%	
	Husband’s refusal	1	0	0	
		4.35%	0.00%	0.00%	
	Medical contraindication	1	0	2	
		4.35%	0.00%	1.52%	
	Vaccine is not available	2	0	1	
		8.70%	0.00%	0.76%	
	Worried about the side effects	16	5	118	
		69.57%	100.00%	89.39%	
COVID-19 vaccine effective	No	4	1	25	0.590
		17.39%	20.00%	18.94%	
	Yes	9	3	72	
		39.13%	60.00%	54.55%	
	Not sure	10	1	35	
		43.48%	20.00%	26.52%	
Comfortable receiving the vaccine	No	3	2	29	0.141
		13.04%	40.00%	21.97%	
	Yes	10	2	77	
		43.48%	40.00%	58.33%	
	Not sure	10	1	26	
		43.48%	20.00%	19.70%	0.217
Source information for the vaccine	Family	1	1	5	
		4.35%	20.00%	3.79%	
	Friends	1	0	1	
		4.35%	0.00%	0.76%	
	Physician	8	2	34	
		34.78%	40.00%	25.76%	
	Relative	2	1	6	
		8.70%	20.00%	4.55%	
	Social media	11	1	86	
		47.83%	20.00%	65.15%	
Fear of receiving the vaccine	COVID-19 is not considered a dangerous disease	0	0	1	0.903
		0.00%	0.00%	0.76%	
	Vaccine will affect the fetus	13	4	84	
		56.52%	80.00%	63.64%	
	Family hesitancy	0	0	4	
		0.00%	0.00%	3.03%	
	Fear of injection	2	0	14	
		8.70%	0.00%	10.61%	
	Not enough studies on pregnant women	8	1	26	
		34.78%	20.00%	19.70%	
	Vaccination may cause infection	0	0	1	
		0.00%	0.00%	0.76%	
	Vaccine effectiveness is low	0	0	2	
		0.00%	0.00%	1.52%	

**Table 7 vaccines-11-00361-t007:** Multinomial logistic regression to estimate the parameters.

Variable		B	Std. Error	Wald	*p*-Value	Odds Ratio
**AstraZeneca Oxford**						
Reason for delaying receiving the vaccine	Intercept	−0.344	0.674	0.260	0.610	
	Allergy	15.229	6330.629	0.000	0.998	4.11 × 10^6^
	Family refusal	1.094	1.656	0.437	0.509	2.986
	Medical advice	−14.547	1129.260	0.000	0.990	0.000
	Medical contraindication	15.212	1088.269	0.000	0.989	4.04 × 10^6^
	Not due	15.045	690.295	0.000	0.983	3.42 × 10^6^
	Vaccine is not available	−0.403	1.106	0.133	0.716	0.669
	Want to take particular type and avoid other types	0.081	1.398	0.003	0.954	1.085
	Worried about the side effects	Ref.				
COVID-19 vaccine effective	No	0.352	1.242	0.080	0.777	1.422
	Not sure	−0.301	1.063	0.080	0.777	0.740
	Yes	Ref.				
Comfortable receiving the vaccine	No	−2.629	1.334	3.884	0.049	0.072
	Not sure	−0.541	1.038	0.272	0.602	0.582
	Yes	Ref.				
Source information for the vaccine	Family	0.065	1.311	0.002	0.961	1.067
	Friends	−0.658	1.559	0.178	0.673	0.518
	Physician	0.543	0.680	0.636	0.425	1.721
	Relative	0.154	1.083	0.020	0.887	1.167
	Social media	Ref.				
Fear of receiving the vaccine	COVID-19 is not considered a dangerous disease	−16.099	5667.250	0.000	0.998	0.000
	Family hesitancy	−12.620	916.167	0.000	0.989	0.000
	Fear of injection	0.884	1.132	0.610	0.435	2.421
	Not enough studies on pregnant females	1.269	0.734	2.985	0.084	3.557
	Vaccination may cause the infection	−16.099	5667.250	0.000	0.998	0.000
	Vaccine effectiveness is low	−13.331	1178.760	0.000	0.991	0.000
	Vaccine will affect the fetus	Ref.				
**Moderna**						
Reason for delaying receiving the vaccine	Intercept	−1.990	1.317	2.283	0.131	
	Allergy	13.386	13,215.981	0.000	0.999	6.51 × 10^5^
	Family refusal	−12.840	6740.206	0.000	0.998	0.000
	Medical advice	−13.780	2141.261	0.000	0.995	0.000
	Medical contraindication	0.555	3492.004	0.000	1.000	1.742
	Not due	0.638	2402.920	0.000	1.000	1.893
	Vaccine is not available	−13.805	1889.628	0.000	0.994	0.000
	Want to take particular type and avoid other types	−11.200	855.777	0.000	0.990	0.000
	Worried about the side effects	Ref.				
COVID-19 vaccine effective	No	−3.329	1.872	3.163	0.075	0.036
	Not sure	−2.841	1.656	2.945	0.086	0.058
	Yes	Ref.				
Comfortable receiving the vaccine	No	1.725	1.727	0.997	0.318	5.610
	Not sure	0.687	1.614	0.181	0.670	1.988
	Yes	Ref.				
Source information for the vaccine	Family	2.294	1.721	1.778	0.182	9.918
Fear of receiving the vaccine	Friends	−12.469	2030.258	0.000	0.995	0.000
	Physician	1.074	1.369	0.615	0.433	2.926
	Relative	2.233	1.662	1.804	0.179	9.323
	Social media	Ref.				
Fear of receiving the vaccine	COVID-19 is not considered a dangerous disease	−14.667	0.000			0.000
	Family hesitancy	−10.251	862.305	0.000	0.991	0.000
	Fear of injection	−11.556	713.774	0.000	0.987	0.000
	Not enough studies on pregnant women	0.573	1.443	0.158	0.691	1.774
	Vaccination may cause infection	−14.667	12,154.897	0.000	0.999	0.000
	Vaccine effectiveness is low	−12.775	2460.592	0.000	0.996	0.000
	Vaccine will affect the fetus	Ref.				
**Pfizer**						
Reason for delaying receiving the vaccine	Intercept	2.502	0.476	27.589	0.000	
	Allergy	28.760	2878.947	0.000	0.992	3.09 × 10^12^
	Family refusal	−15.492	1765.371	0.000	0.993	0.000
	Medical advice	−1.540	1.357	1.288	0.256	0.214
	Medical contraindication	14.089	1088.268	0.000	0.990	1.31 × 10^6^
	Not due	13.249	690.295	0.000	0.985	5.67 × 10^5^
	Vaccine is not available	−3.699	1.319	7.865	0.005	0.025
	Want to take particular type and avoid other types	−0.147	1.047	0.020	0.889	0.864
	Worried about the side effects	Ref.				
COVID-19 vaccine effective	No	−0.621	0.833	0.556	0.456	0.537
	Not sure	−0.726	0.709	1.050	0.306	0.484
	Yes	Ref.				
Comfortable receiving the vaccine	No	−1.351	0.824	2.687	0.101	0.259
	Not sure	−1.078	0.722	2.233	0.135	0.340
	Yes	Ref.				
Source information for the vaccine	Family	−0.725	0.857	0.716	0.397	0.484
	Friends	−2.169	1.405	2.384	0.123	0.114
	Physician	−0.222	0.481	0.212	0.645	0.801
	Relative	−0.935	0.818	1.305	0.253	0.393
	Social media	Ref.				
Fear of receiving the vaccine	COVID-19 is not considered a dangerous disease	−16.235	2365.644	0.000	0.995	0.000
	Family hesitancy	1.463	1.517	0.929	0.335	4.319
	Fear of injection	0.534	0.856	0.389	0.533	1.706
	Not enough studies on pregnant women	0.486	0.521	0.869	0.351	1.625
	Vaccination may cause infection	−16.235	2365.644	0.000	0.995	0.000
	Vaccine effectiveness is low	−0.988	1.154	0.733	0.392	0.372
	Vaccine will affect the fetus	Ref.				

(Table 7 to be kept as E-supplement), *p* value less than 0.05 was considered statistically significant.

## Data Availability

Not applicable.

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
