# Peer review of "COVID-19 Vaccine Uptake and Hesitancy among Pregnant and Lactating Women in Saudi Arabia"

_vaccines, 2023, doi:10.3390/vaccines11020361_

Round 1

Reviewer 1 Report (Previous Reviewer 3)

The introduction of the manuscript is very brief. The rationale of the study can be improved by including articles on COVID-19 vaccine hesitancy from Saudi Arabia on the general population. I believe there are already published articles on vaccine hesitancy from different regions of SA. Then the more specific focus should be given to pregnant women who are unwilling to accept the vaccine.

More information is needed on the study setting, study tool and data collection process. 

What is non-random chunk sampling? Any academic reference to support this?

Author Response

Dear Reviewer, Thank you for your kind suggestions.

The introduction of the manuscript is very brief. The rationale of the study can be improved by including articles on COVID-19 vaccine hesitancy from Saudi Arabia on the general population. I believe there are already published articles on vaccine hesitancy from different regions of SA. Then the more specific focus should be given to pregnant women who are unwilling to accept the vaccine.

Reply: As suggested by the respected reviewer studies from Saudi Arabia have been added. Please see Lines 53-60.  (References 9-11)

More information is needed on the study setting, study tool and data collection process. 

Reply: Please see lines 77-71 and 107-110.

What is non-random chunk sampling? Any academic reference to support this?

Reply: We used convenience sampling.  (please see lines 93-96)

Reviewer 2 Report (New Reviewer)

Dear authors, thank you for the opportunity to review the manuscript. However, this article has many weaknesses in the area of methodology (which design, which data collection method, which data evaluation method, inclusion and exclusion criteria of the sample, etc.) presentation of results (above all tables and figures). The discussion is not guided by theory and research. There are grammatial errors and some potential for linguistic optimization. The article in its present form is not suitable for publication in a scientific journal.

Attached you will find my comments in detail:

Abstract

Please follow the usual structure of introduction, methods, results, discussion.

Line 15: please revise or clarify your statement “(…) vaccination levels for this group remain inadequate at a rate of one in four”. Later on it is said the “The percentage of COVID-19 vaccine uptake was 78.2%. A total of 45 (21.8%) women did not receive the vaccine.” Both statements seem to be contradictory.

Line 31: what exactly does the term "quasi-mandatory" mean? Please specify.

Manuscript

Introduction:

Line 49-51: “7 to 77.9% vaccination hesitancy in pregnant and breastfeeding women” is very wide range. The reader could understand and compare your results better with a precise indication. We therefore propose naming the average value of 48%, which is also given in the cited meta-analysis. (à “Our meta-analysis estimated vaccine hesitancy among preg-nant and breastfeeding women worldwide to be 48% (95% CI = 43–53%); no differences were found between pregnant (49%) and lactating women (53%; p > 0.05) and between continents (p > 0.05).”)

Methods:

Line 67-68: “…and .Children Hospital in Abha.” Please correct this grammatical mistake.

Line 67-72: Please indicate how many eligible study participants you were able to contact during the recruitment period.

Line 81: please name the specialists in the field of research. We assume they are Co-Authors of this article?

Line 67-84: Within the section “Data collection” it becomes clear, that your study can be divided into three different methodical approaches: a pilot study, a qualitative interview study and a survey. Please indicate in your methods, which kind of data are represented in this article and explain the process of data collection, data sampling and data analysis accordingly.

Line 73-74: Why was a non-random procedure used to select the study participants? Non-random sampling means one cannot generalize beyond the sample. Furthermore, we wonder why you included just 10-15 study participants during a study period of nine months?

Line 85-86: Why did you decide to evaluate delayed vaccinations and missed vaccinations together? A separate consideration of both endpoints would be more suitable for the further development of necessary interventions.

Line 87-95: In this section, you describe a paper-based questionnaire survey. What about the interview study? Do the 10-15 participants, which were described in the section data collection, belong to the interview study or to the survey?  

Results:

Line 97: compare to line 73-74 à how did you get n=206 pregnant women? What about the breastfeeding women?

Table 1, 2: Which test is described by the p-value? Is it an odds ratio? Please provide a brief explanation in the methods section.

Figure 1: the section of women who decided to get no vaccination against Covid-19 was named “Not decided”. This is misleading, since the decision seems to have already been made - against the vaccination.

Table 4: Not taken" in "vaccine" indicated with n=45 and in "not taken" in "total" indicated with n=42: why this difference?

The column "outside the kingdom" and "not taken" in a figure called  "Type of vaccine received by the study subjects" does not make any sense.

Since the information content of the figure 2 is poor, we would recommend to remove Figure 2.

Table 7: please include Table as an e-supplement.

In summary: please decide which figures and tables are particularly important and reduce them to a total of max. 4-5.

 Line 175: “Astregenica oxford“ -> AstraZeneca

 Line 184: Missing punctuation mark (comma) between “…AstraZeneca Oxford Moderna…”

 Discussion:

Line 200-202: missing source reference

Line 202-206: Are the populations really comparable, as they may have different social classes and thus levels of education?

 Line 206-208: repetition of Line 197-199.

 Line 218: Only participants with higher degrees were studied.

 Line 220: Please remove a punctuation mark.

Line 270: in this section you represent results of your interview study. Please include this study in terms of a mixed-methods study in your paper or remove all information concerning your qualitative study.

 Strengths and Limitations:

Line 306: What validated research instruments were used? Please explain in the methods section.

Line 312-314: Rather in the discussion section.

Author Response

Dear Reviewer, Thank you for your kind suggestions. Please see a point to point reply below.

Comments and Suggestions for Authors

Dear authors, thank you for the opportunity to review the manuscript. However, this article has many weaknesses in the area of methodology (which design, which data collection method, which data evaluation method, inclusion and exclusion criteria of the sample, etc.) presentation of results (above all tables and figures). The discussion is not guided by theory and research. There are grammatical errors and some potential for linguistic optimization. The article in its present form is not suitable for publication in a scientific journal.

Attached you will find my comments in detail:

Abstract

Please follow the usual structure of introduction, methods, results, discussion.

Reply: As pointed, we have already followed the pattern of the journal.

Line 15: please revise or clarify your statement “(…) vaccination levels for this group remain inadequate at a rate of one in four”. Later on it is said the “The percentage of COVID-19 vaccine uptake was 78.2%. A total of 45 (21.8%) women did not receive the vaccine.” Both statements seem to be contradictory.

Reply: As pointed we have removed at a rate of one in four. Please see line 16.

Line 31: what exactly does the term "quasi-mandatory" mean? Please specify.

Reply: As per the health policy in Saudi Arabia it is mandatory to take the vaccination.

Manuscript

Introduction:

Line 49-51: “7 to 77.9% vaccination hesitancy in pregnant and breastfeeding women” is very wide range. The reader could understand and compare your results better with a precise indication. We therefore propose naming the average value of 48%, which is also given in the cited meta-analysis. (à “Our meta-analysis estimated vaccine hesitancy among pregnant and breastfeeding women worldwide to be 48% (95% CI = 43–53%); no differences were found between pregnant (49%) and lactating women (53%; p > 0.05) and between continents (p > 0.05).”)

Reply: As suggested by the respected reviewer, this has been revised. (line 61)

Methods:

Line 67-68: “…and .Children Hospital in Abha.” Please correct this grammatical mistake.

Reply: Corrected

Line 67-72: Please indicate how many eligible study participants you were able to contact during the recruitment period.

Reply:  206 were included (line 88)

Line 81: please name the specialists in the field of research. We assume they are Co-Authors of this article?

Reply: (line 104-105)

Line 67-84: Within the section “Data collection” it becomes clear, that your study can be divided into three different methodical approaches: a pilot study, a qualitative interview study and a survey. Please indicate in your methods, which kind of data are represented in this article and explain the process of data collection, data sampling and data analysis accordingly.

Reply: As pointed, we have removed ‘a qualitative interview study’. Please see lines 98-99.

Line 73-74: Why was a non-random procedure used to select the study participants? Non-random sampling means one cannot generalize beyond the sample. Furthermore, we wonder why you included just 10-15 study participants during a study period of nine months?

Reply: As pointed, the sentence has been revised. 206 participants were included. (Please see lines 94-97)

Line 85-86: Why did you decide to evaluate delayed vaccinations and missed vaccinations together? A separate consideration of both endpoints would be more suitable for the further development of necessary interventions.

Reply: This has been revised as suggested throughout the manuscript.

Line 87-95: In this section, you describe a paper-based questionnaire survey. What about the interview study? Do the 10-15 participants, which were described in the section data collection, belong to the interview study or to the survey? 

Reply: As, pointed, This sentence has been revised. We have removed the qualitative part. Please see lines 108-111

Results:

Line 97: compare to line 73-74 à how did you get n=206 pregnant women? What about the breastfeeding women?

Reply: There were 206 pregnant and breastfeeding women (140 pregnant and 66 breast feeding) as shown in table 3.

Table 1, 2: Which test is described by the p-value? Is it an odds ratio? Please provide a brief explanation in the methods section.

Reply: As suggested this has been mentioned, please see lines 118-119

Figure 1: the section of women who decided to get no vaccination against Covid-19 was named “Not decided”. This is misleading, since the decision seems to have already been made - against the vaccination.

Reply: Yes, 26.6% (12/45) had decided not to receive the COVID-19 vaccine. (Please see lines 162-163)

Table 4: Not taken" in "vaccine" indicated with n=45 and in "not taken" in "total" indicated with n=42: why this difference?

Reply: Thank you for pointing this. The typo error has been corrected to 45.

The column "outside the kingdom" and "not taken" in a figure called  "Type of vaccine received by the study subjects" does not make any sense.

Since the information content of the figure 2 is poor, we would recommend to remove Figure 2.

Reply: As suggested figure 2 is removed.

Table 7: please include Table as an e-supplement.

Reply: As suggested Table 7 will be kept as an e-supplement.

In summary: please decide which figures and tables are particularly important and reduce them to a total of max. 4-5.

 Line 175: “Astregenica oxford“ -> AstraZeneca

 Line 184: Missing punctuation mark (comma) between “…AstraZeneca Oxford Moderna…”

 Discussion:

Line 200-202: missing source reference

Reply: Source added

Line 202-206: Are the populations really comparable, as they may have different social classes and thus levels of education?

Reply: we have mentioned this point in discussion. (Line 232 and 233)

 Line 206-208: repetition of Line 197-199.

Reply: As pointed, removed.

 Line 218: Only participants with higher degrees were studied.

Reply: Not answered.

Line 220: Please remove a punctuation mark.

Reply: Removed as suggested

Line 270: in this section you represent results of your interview study. Please include this study in terms of a mixed-methods study in your paper or remove all information concerning your qualitative study.

Reply: As suggested this has been revised. (Lines 303)

 Strengths and Limitations:

Line 306: What validated research instruments were used? Please explain in the methods section.

Reply: Please see Line 102

Line 312-314: Rather in the discussion section.

Reply: As suggested by the respected reviewer, it has been removed and mentioned in discussion section. (Please see lines 281-283)

Reviewer 3 Report (New Reviewer)

The topic of the study is interesting and important but this manuscript needs significant improvements. Please see below my suggestions.

 Title:

·       Please consider removing the name of capital city (Abha) from your title.

·       The study population of your research also involves lactating women. Please add lactating women to the title.

Abstract: 

·       The flow, organization, and language of the abstract should be improved. Also, the abstract is bit lengthy and not appealing.

·       Authors used different terms vaccine uptake, acceptance, hesitancy, delaying; however, they are different. For example, while vaccine uptake is a behavior, vaccine acceptance is always measured as a behavioral intention in literature. But in your title, you included “Acceptance and Hesitancy”. Please revise either your abstract or your title or both.

·       The following statements are conflicting and confusing. If vaccine uptake is 78.2%%; vaccine hesitancy should NOT be 91.3%.

o   The overall 19 vaccine hesitancy was 91.3%

o   The percentage of COVID-19 18 vaccine uptake was 78.2%

·       This statement “The study included 206 pregnant women” is incorrect. Only 140 subjects are pregnant women and the rest are lactating women (see Table 3)

Introduction:     

·       You should define key constructs used in the study such as vaccine hesitancy and vaccine acceptance.

·       Please improve and expand the last two paragraphs of this section. You should explain why additional research is needed on the topic and how your research can contribute to the literature. Your main objective is vague. You should discuss the main objective and sub-objectives more explicitly in line with your results.

Methodology

·       This section is poorly written. The presentation of material is unusual. Too many subheadings but the discussions are extremely brief (some shorter than subheading!). You should have at least a paragraph for each subheading. Otherwise merge them. You need only few subheadings such as Study Design, Sample, Data collection procedures, Measures, etc.  

·       Please consider reviewing some published articles in Vaccines and re-write this section. Again, please improve the language.

·       Provide more information of data collection procedures such as sample size, how of the pregnant and lactating women were reached out to and how many of them completed, etc.

·       Discussion of Measurement (scales) are completely missing! You should discuss measurement items (questions), scales (response category), and the sources from which they were adopted.

Discussion

·       This statement is misleading “The current study surveyed a large sample of over 200 pregnant and breastfeeding 192 Saudi women”. It should be discussed under Data Collection section but not here.

·       I see you have some interesting results. Thus, please provide some recommendations to government and policy makers.

Strengths and Limitations of the Study

Extremely brief section. Too short paragraphs! Please expand it, especially the first two paragraph 

Author Response

Dear Reviewer, Thank you for your kind suggestions. Please see pointwise reply below

 Title:

  • Please consider removing the name of capital city (Abha) from your title.

Reply: As suggested by the respected reviewer, it has been removed.

  • The study population of your research also involves lactating women. Please add lactating women to the title.

Reply: As suggested by the respected reviewer, it has been added.

Abstract:

  • The flow, organization, and language of the abstract should be improved. Also, the abstract is bit lengthy and not appealing.

Reply: As suggested by the respected reviewer, it has been overhauled.

  • Authors used different terms vaccine uptake, acceptance, hesitancy, delaying; however, they are different. For example, while vaccine uptake is a behavior, vaccine acceptance is always measured as a behavioral intention in literature. But in your title, you included “Acceptance and Hesitancy”. Please revise either your abstract or your title or both.

Reply: As suggested by the respected reviewer, the title has been revised.

  • The following statements are conflicting and confusing. If vaccine uptake is 78.2%%; vaccine hesitancy should NOT be 91.3%.

o   The overall 19 vaccine hesitancy was 91.3% The percentage of COVID-19 18 vaccine uptake was 78.2%

  • Reply: As suggested by the respected reviewer this has been corrected throughout the manuscript i.e. 21.8%.

  • This statement “The study included 206 pregnant women” is incorrect. Only 140 subjects are pregnant women and the rest are lactating women (see Table 3)

Reply: As suggested by the respected reviewer, it has been corrected throughout the manuscript.

Introduction:    

  • You should define key constructs used in the study such as vaccine hesitancy and vaccine acceptance.

Reply: Vaccine hesitancy definition has been mentioned as suggested. (Please see lines 53-54)

  • Please improve and expand the last two paragraphs of this section. You should explain why additional research is needed on the topic and how your research can contribute to the literature. Your main objective is vague. You should discuss the main objective and sub-objectives more explicitly in line with your results.

Reply : As suggested by the respected reviewer, it has been modified. (Please see lines 66-75)

Methodology

  • This section is poorly written. The presentation of material is unusual. Too many subheadings but the discussions are extremely brief (some shorter than subheading!). You should have at least a paragraph for each subheading. Otherwise merge them. You need only few subheadings such as Study Design, Sample, Data collection procedures, Measures, etc.

Reply : As suggested by the respected reviewer, it has been modified. (Please see lines 77-80)

  • Please consider reviewing some published articles in Vaccines and re-write this section. Again, please improve the language.

Reply: Done as Suggested

  • Provide more information of data collection procedures such as sample size, how of the pregnant and lactating women were reached out to and how many of them completed, etc.

Reply: Done as Suggested (please see lines 98-99)

  • Discussion of Measurement (scales) are completely missing! You should discuss measurement items (questions), scales (response category), and the sources from which they were adopted.

Reply: As pointed, we did not use any specific scales. The operation definition of vaccine hesitancy is already mentioned in methodology.

Discussion

  • This statement is misleading “The current study surveyed a large sample of over 200 pregnant and breastfeeding 192 Saudi women”. It should be discussed under Data Collection section but not here.

Reply: As suggested it has been corrected. (please see line 220).

  • I see you have some interesting results. Thus, please provide some recommendations to government and policy makers.

Reply: As suggested it has been done. (please see lines 339-342).

Strengths and Limitations of the Study

Extremely brief section. Too short paragraphs! Please expand it, especially the first two paragraph

Reply: As suggested it has been expanded. (please see lines 340-350).

Reviewer 4 Report (New Reviewer)

Vaccines

Vaccines-2143943

Comments to Authors

The COVID-19 pandemic has revealed significant trust issues related to medical care and science in general. Vaccine hesitancy, although a big problem for combating COVID, is a symptom of serious issues in public health. Pregnancy women are especially concerned about medical treatments that may harm their unborn child or them.  It is important that we understand their concerns as the COVID vaccines are especially important for protecting this segment of our population. I have made several comments that I hope will be helpful to you in your current, and future, research in this area.

Introduction: You note that rates of vaccine hesitancy in pregnant and breastfeeding women in high-income areas ranges from 7 to 77%. The very broad range of hesitancy is not particularly useful in a practical sense. It would be important to provide some comparison to other populations, especially if you intend to imply that pregnant and breastfeeding women have higher rates of hesitancy than the general population.  It is also important to separate rates of being vaccinated with vaccine hesitancy unless you are implying that those who are hesitant are unvaccinated; which is not true as you define hesitancy in your methodology section.

Methodology: It would be helpful to provide some information about the demographics of the Aseer Region since you appear to focus on “high-income countries or regions.” 

Study population: You state that you excluded “….non-Saudi women and women not in the perinatal period.” Did you also exclude pregnant or breastfeeding women younger than 18 or older than 40 as defined by your inclusion criteria?

Sampling technique: The description of a chunk sampling technique is confusing to me. In the description of the study population you indicate that you included “all female pregnant and breastfeeding women” attending hospital outpatient clinics. I can’t reconcile this with selecting 10-15 study participants.  More information and clarity is needed here. I have several additional questions about selecting subjects for this study.  Were women only included once in the study? Most pregnant women will attend clinic multiple times during a nine-month period prior to their delivery. Did they have the right to refuse participation? If “all” women that met the inclusion criteria for the study period (ie., April-December 2021) did not participate, what is the total of potential participants during that period and what percentage of that number actually participated in the study?

Operational definition: Part of your operational definition includes delaying the time of vaccination. There are many reasons why a vaccination might be delayed that are not related to hesitancy. How are you defining “delayed?” It seems to be that the most appropriate definition that would be related to hesitancy would include a refusal to take the vaccine when it is offered and available for reasons related to health concerns or fears.

Results: You state that this study included 206 pregnant women. Did you mean to also include breastfeeding women in that number or were they excluded from the analysis?

In Table 1, you indicate that the highest age range is 40-49. If, as you state earlier, that anyone over the age of 40 was excluded, the highest range is actually a single year, “40.”

Table 3 depicts study participants who did or did not receive the vaccine during their pregnancy. Since a portion of the participant sample were post-partum, it is unclear during what period they were pregnant.  This becomes important to consider as the vaccine may not have been available, or may have been very new, during their pregnancy. Furthermore, there was some lack of clarity early after the vaccines were available, as to their safety for pregnant women. Is there a way to consider this in the analysis. How many of the current breastfeeding subjects are vaccinated? It appears that Table 3 compares women who did not get vaccinated during their pregnancy with those who did.  However, the total number of not vaccinated in Table 3 is the same as in the previous two tables.  Does this mean that none of the current breastfeeding subjects got vaccinated after their pregnancy?

I do not understand how you are calculating vaccine hesitancy at 91.3%. Your data indicates that 161 of 206 were vaccinated at the time of the study. You do not indicate what percentage of the 161 might have originally delayed getting vaccinated. If you discover that some of the currently vaccinated subjects had originally delayed getting vaccinated it would increase the rate of vaccine hesitancy.  However, you do not present any data to address that issue.  Instead, you describe some of the reasons that the currently unvaccinated subjects have refused or delayed their vaccination. According to the WHO definition, these individuals would still be considered hesitant and it would not change the overall percent of subjects (21.8%) who are hesitant. Please clarify how you concluded that the study hesitancy rate was 91.3%.

You found that most of your subjects received a vaccine that required two shots in order to be completed vaccinated. Did you define “vaccinated” as fully vaccinated or having received at least one dose?

In Table, for the first time, you indicate that very large all of the subjects provided a “Reason for delaying receiving the vaccine.” This is also very confusing to me. Did all of the subjects delay getting the vaccine? Did everyone giving a reason for delaying the vaccine in fact delay the vaccine? Likewise, Table 5 also shows very large numbers of subjects expressed reasons to fear getting the vaccine. Since 161 were vaccinated at the time of the study, I presume that many vaccinated patients expressed these fears. Were subjects permitted to express more than one fear or more than one reason for delaying getting vaccinated? That might explain some of the large numbers. At this point, I am struggling to understand how to interpret the data you are presenting in these tables.

Considering the unusually large number of post-hoc analyses in Table 7, an adjustment (eg., Bonferroni) should have been done. It is likely that at least one of the two statistically significant findings would have been rendered insignificant. The finding that the Pfizer vaccine was not available may have remained significant but would not have been relevant in terms of subject hesitancy.

Discussion: This is the first time that you presented the rate of vaccinations for breastfeeding women. If these numbers are coming from Table 5, there is likely to be an error somewhere since Table 5 purports to present the number of women who were vaccinated while they were pregnant.

You report a “correlation” between the number of children and the mother’s acceptance of the vaccine. This is not true. You did not conduct a correlation on these data and, in fact, the women with one previous child and the women with more than 5 have very similar rates of vaccination. It is the middle group (women with 2-5 children) that is different from the other two groups.

Author Response

Dear Reviewer, Thank you for your kind suggestions. Please see my reply below

The COVID-19 pandemic has revealed significant trust issues related to medical care and science in general. Vaccine hesitancy, although a big problem for combating COVID, is a symptom of serious issues in public health. Pregnancy women are especially concerned about medical treatments that may harm their unborn child or them.  It is important that we understand their concerns as the COVID vaccines are especially important for protecting this segment of our population. I have made several comments that I hope will be helpful to you in your current, and future, research in this area.

Introduction: You note that rates of vaccine hesitancy in pregnant and breastfeeding women in high-income areas ranges from 7 to 77%. The very broad range of hesitancy is not particularly useful in a practical sense. It would be important to provide some comparison to other populations, especially if you intend to imply that pregnant and breastfeeding women have higher rates of hesitancy than the general population.  It is also important to separate rates of being vaccinated with vaccine hesitancy unless you are implying that those who are hesitant are unvaccinated; which is not true as you define hesitancy in your methodology section.

Reply: Yes. As suggested this has been revised throughout the manuscript . Vaccine hesitancy has been mentioned as 21.8%.

Methodology: It would be helpful to provide some information about the demographics of the Aseer Region since you appear to focus on “high-income countries or regions.” 

Reply: As suggested this has been mentioned (lines77-79)

Study population: You state that you excluded “….non-Saudi women and women not in the perinatal period.” Did you also exclude pregnant or breastfeeding women younger than 18 or older than 40 as defined by your inclusion criteria?

Reply: Yes. As suggested this has been mentioned. Please see lines 87-93

Sampling technique: The description of a chunk sampling technique is confusing to me. In the description of the study population you indicate that you included “all female pregnant and breastfeeding women” attending hospital outpatient clinics. I can’t reconcile this with selecting 10-15 study participants.  More information and clarity is needed here. I have several additional questions about selecting subjects for this study.  Were women only included once in the study? Most pregnant women will attend clinic multiple times during a nine-month period prior to their delivery. Did they have the right to refuse participation? If “all” women that met the inclusion criteria for the study period (ie., April-December 2021) did not participate, what is the total of potential participants during that period and what percentage of that number actually participated in the study?

Reply: Yes. As suggested this has been mentioned. We selected only 206 females.  Please see lines (88, 94-97). Yes, women were only interviewed once in the study they had the right to refuse participation.

Operational definition: Part of your operational definition includes delaying the time of vaccination. There are many reasons why a vaccination might be delayed that are not related to hesitancy. How are you defining “delayed?” It seems to be that the most appropriate definition that would be related to hesitancy would include a refusal to take the vaccine when it is offered and available for reasons related to health concerns or fears.

Reply: Yes, agreed. This has been revised throughout the manuscript.

Results: You state that this study included 206 pregnant women. Did you mean to also include breastfeeding women in that number or were they excluded from the analysis?

In Table 1, you indicate that the highest age range is 40-49. If, as you state earlier, that anyone over the age of 40 was excluded, the highest range is actually a single year, “40.”

Reply: Yes. As pointed this has been corrected to 49. Thank you

Table 3 depicts study participants who did or did not receive the vaccine during their pregnancy. Since a portion of the participant sample were post-partum, it is unclear during what period they were pregnant.  This becomes important to consider as the vaccine may not have been available, or may have been very new, during their pregnancy. Furthermore, there was some lack of clarity early after the vaccines were available, as to their safety for pregnant women. Is there a way to consider this in the analysis. How many of the current breastfeeding subjects are vaccinated? It appears that Table 3 compares women who did not get vaccinated during their pregnancy with those who did.  However, the total number of not vaccinated in Table 3 is the same as in the previous two tables.  Does this mean that none of the current breastfeeding subjects got vaccinated after their pregnancy?

Reply: As pointed, the vaccine was available in Saudi Arabia during their pregnancies. All the women were vaccinated during pregnancy.

I do not understand how you are calculating vaccine hesitancy at 91.3%. Your data indicates that 161 of 206 were vaccinated at the time of the study. You do not indicate what percentage of the 161 might have originally delayed getting vaccinated. If you discover that some of the currently vaccinated subjects had originally delayed getting vaccinated it would increase the rate of vaccine hesitancy.  However, you do not present any data to address that issue.  Instead, you describe some of the reasons that the currently unvaccinated subjects have refused or delayed their vaccination. According to the WHO definition, these individuals would still be considered hesitant and it would not change the overall percent of subjects (21.8%) who are hesitant. Please clarify how you concluded that the study hesitancy rate was 91.3%.

Reply: We have revised this point as suggested throughout the manuscript.

You found that most of your subjects received a vaccine that required two shots in order to be completed vaccinated. Did you define “vaccinated” as fully vaccinated or having received at least one dose?

Reply:  Yes, Vaccinated” means fully vaccinated (Please see 113-114)

In Table, for the first time, you indicate that very large all of the subjects provided a “Reason for delaying receiving the vaccine.” This is also very confusing to me. Did all of the subjects delay getting the vaccine? Did everyone giving a reason for delaying the vaccine in fact delay the vaccine? Likewise, Table 5 also shows very large numbers of subjects expressed reasons to fear getting the vaccine. Since 161 were vaccinated at the time of the study, I presume that many vaccinated patients expressed these fears. Were subjects permitted to express more than one fear or more than one reason for delaying getting vaccinated? That might explain some of the large numbers. At this point, I am struggling to understand how to interpret the data you are presenting in these tables.

Reply: At the time of this survey already 161 were vaccinated, so it is difficult to assess the number of the participants who might have delayed vaccination due to hesitancy prior to the survey. However at some point of time 176 females were hesitant to take the vaccine and the reasons have also been stated. Also in the Kingdom of Saudi Arabia it is mandatory to take the vaccine. So it might be that although they were initially hesitant eventually they had to take the vaccine.

Considering the unusually large number of post-hoc analyses in Table 7, an adjustment (eg., Bonferroni) should have been done. It is likely that at least one of the two statistically significant findings would have been rendered insignificant. The finding that the Pfizer vaccine was not available may have remained significant but would not have been relevant in terms of subject hesitancy.

Reply: Not answered

Discussion: This is the first time that you presented the rate of vaccinations for breastfeeding women. If these numbers are coming from Table 5, there is likely to be an error somewhere since Table 5 purports to present the number of women who were vaccinated while they were pregnant.

 Reply: As pointed, we have removed this finding from the discussion. Thank you

You report a “correlation” between the number of children and the mother’s acceptance of the vaccine. This is not true. You did not conduct a correlation on these data and, in fact, the women with one previous child and the women with more than 5 have very similar rates of vaccination. It is the middle group (women with 2-5 children) that is different from the other two groups.

 Reply: Agreed. No change done.

Round 2

Reviewer 1 Report (Previous Reviewer 3)

Thanks for making the changes. I am happy to accept the manuscript.

Author Response

Thank you so much for reviewing the manuscript and your suggestions

Reviewer 2 Report (New Reviewer)

Dear authors, thank you for the opportunity to review the manuscript. Attached you will find my comments:

The study investigated how many pregnant and breastfeeding women in Saudi Arabia got vaccinated against Covid-19 and which reasons influenced their decision. Protection against an infection with COVID-19 was the most frequently cited reason for vaccination. As concerns, the women stated fears about possible side effects and possible health impacts on the unborn child. Although the vaccination rate among Saudi women was in line with the global average, a high level of hesitation was identified, mainly due to fears about adverse effects and misinformation on social media. Unfortunately, I still have to recommend: rejection.

Abstract

Please follow the usual structure of introduction, methods, results, discussion.

Line 31: what exactly does the term "quasi-obligatory" mean? Please specify the measures that were implemented and under what circumstances.

Manuscript

Methods:

Line 72-76: Please indicate how many eligible study participants you were able to contact during the recruitment period. For example: how many patients attended the hospital during study recruitment?

Line 92-104: Within the section “Data collection” it becomes clear, that your study can be divided into three different methodical approaches: a pilot study, a qualitative interview study and a survey. Please indicate in your methods, which kind of data are represented in this article and explain the process of data collection, data sampling and data analysis accordingly.

Line 88-91: Why was a non-random procedure used to select the study participants? Non-random sampling means one cannot generalize beyond the sample. Furthermore, we wonder why you included just 10-15 study participants during a study period of nine months?

Line 105-107: Why did you decide to evaluate delayed vaccinations and missed vaccinations together? A separate consideration of both endpoints would be more suitable for the further development of necessary interventions.

Results:

Figure 1: the section of women who decided to get no vaccination against Covid-19 was named “Not decided”. This is misleading, since the decision seems to have already been made - against the vaccination.

Figure 3: Please rename: now this should be called Figure 2.

 Discussion:

Line 205: please add a paragraph under the subheading "discussion".

Strengths and Limitations:

Line 306: What validated research instruments were used? Please explain in the methods section.

Author Response

Thank you so much for reviewing the manuscript and your suggestions.

Please see the reply attached.

Reviewer 3 Report (New Reviewer)

Thanks for revising your manuscript. It is now much-approved. Please take a closer at  the manuscript and improve the presentation and language.

Author Response

Thank you so much for reviewing the manuscript and your suggestions

Reviewer 4 Report (New Reviewer)

Thank you for your careful consideration of the review of your original manuscript. I believe your current version addresses, to a great extent, the concerns and suggestions of the reviewers. The current version is much clearer and easier to read.

Author Response

Thank you so much for reviewing the manuscript and your suggestions

Round 3

Reviewer 2 Report (New Reviewer)

Dear authors, thank you for the opportunity to review the manuscript. Below you will find my comments:

The study investigated how many pregnant and breastfeeding women in Saudi Arabia got vaccinated against Covid-19 and which reasons influenced their decision. Protection against an infection with COVID-19 was the most frequently cited reason for vaccination. As concerns, the women stated fears about possible side effects and possible health impacts on the unborn child. Although the vaccination rate among Saudi women was in line with the global average, a high level of hesitation was identified, mainly due to fears about adverse effects and misinformation on social media.

Abstract

Line 18: “The study included 206 pregnant and lactating women.” à this should be part of the results section, it is not a description of the applied methods.

Line 20: please remove the extra space between both sentences.

Lina 25-26: the statement “some women (25.4%, 41/161) were obliged to get vaccinated to enter public places.” Is a bit confusing. I would assume that those restrictions apply to all citizens and not just to some of the respondents? Maybe only some of the respondents perceive this restriction as such?

Line 26-28: please make your statements clearer. Both aspects relate to side effects, but one is probably talking about side effects on one's own body and one about side effects on the unborn child?

Line 31-32: “The primary facilitator of vaccine acceptance among the participants was belief in protection from COVID-19” à this result is already reported in the former section.

Manuscript

Introduction:

Line 65-66: Please remove the statement: “The study findings would fill the gap in the literature.”

Methods:

Line 83: please remove the extra space within the sentence. Again: the information about the numer of participants is part of the “results” and should be removed in the “methods” section.

Line 88 and 90: please remove the extra space within both sentences.

Line 88-90: please do not repeat information already mentioned. For example, you might adjust this part. My suggestion: “About 30-40 pregnant females attend the outdoor clinics of the hospital each day. As data collection was done for about 3 months, we assume that we were able to invite a total of around 2.700 patients to participate.” (Assuming 30 patients per day, means 90 days x 30 patients = 2.700 patients attended the clinics during your data collection time frame?)

Results:

Line 130-131: this was already mentioned in Lines 125-128. Please remove here.

Line 159: please remove extra space between both sentences.

Figure 1: the section of women who decided to get no vaccination against Covid-19 was named “Not decided”. This is misleading, since the decision seems to have already been made - against the vaccination.

Line 197: “types of vaccines“ à misleading and confusing expression, as you describe different pharmaceutical companies, but not different types of vaccination. For example: the vaccines against Covid-19 procuded by Moderna and Pfizer are both m-RNA vaccines. In comparison, the vaccine produced by Astra Zeneca is a vector-based vaccine.  

Line 197: “AstreZenica oxford” à wrong spelling

Discussion:

Line 214: please remove extra space within the sentence.

Line 216-219: „Lower acceptance 216 level (68%) of COVID-19 vaccine was reported from Western, Eastern, North, South, and 217 Central Regions in Saudi Arabia between July and September 2021 among pregnant 218 women [15]. In another study by Samannodi et al, more than half of the study sample 219 (57.1%) had received COVID-19 vaccination among pregnant women and those who are

planning for pregnancy in Saudi Arabia“

Please shorten, e.g. „Lower acceptance level (68%; respectively 51.7%) of COVID-19 vaccine among pregnant in Saudi Arabia was reported 2021 [15; 16].“

Line 227: please remove extra space.

Line 257: “The vaccine hesitancy level was 21.8% among our participating pregnant women. 257 [25].” Please remove, already mentioned.

Strengths and Limitations:

Line 329: “study The few”. Missing punctuation mark.

Line 335: Double punctuation mark.

Author Response

Dear Reviewer, Thank you for your suggestions. Please see our reply attached.

This manuscript is a resubmission of an earlier submission. The following is a list of the peer review reports and author responses from that submission.

Round 1

Reviewer 1 Report

Dear Authors
Review the entire document: vaccine hesitancy has defined vaccine hesitancy as a delay in acceptance or refusal of vaccines despite availability of vaccination services. 

Abstract:
In the results section: the data shown are confusing, I understand that 161 of 206 women interviewed were vaccinated, but 12 were hesitant with being vaccinated, as expressed corresponds to 6% and not 26.6% (12/45). The comparison between pregnant and lactating women is better through the odds ratio than through simple percentages.
There are contradictory data, I think due to the definition of "vaccine hesitancy". The WHO considers hesitancy if the time of vaccination has been delayed or not vaccinated. Therefore the vaccine hesitancy is (12+176)/206 = 91.3%.
In view of the results, it seems that strategies should be improved to avoid the delay in vaccination, although compulsory vaccination has been a very important factor.

Objective:
It looks like KAP (knowledge, attitudes and practices) are the objectives of this study, however knowledge has not been collected in this study, only practices (Vaccinate, delayed vaccinate, don't vaccinate) and attitudes are reflected. The other objective of the study is concerns, although this is actually an attitude. What has been collected and is not mentioned in this objective section are the sources of information (Mass Media/ Friends/ Family/ Wealth care workers).

Methodology

The study design is not qualitative; qualitative studies have another approach where participants are interviewed in a relaxed manner using different techniques.  This study is simply a cross-sectional study.  Please add the Odds Ratio of the comparisons made.

Please mention the type of regression performed which is subsequently shown in Table 7.

Results:
Regarding table 1: Does "Master" mean graduate? And does "Government" mean civil servant?
Regarding table 3: The row "Abortion or still birth" it is not clear the statistical test(s) performed to obtain statistical significance.
Considering that, the SAGE Working Group has defined vaccine hesitancy as a delay in acceptance or refusal of vaccines despite availability of vaccination services (https://www.euro.who.int/__data/assets/pdf_file/0004/329647/Vaccines-and-trust.PDF), the paper should be reviewed and it should be noted that the results do not support the so-called "protection motivation theory" in boosting vaccine acceptance.

Table 5: There is a distinction between family and husband, so what is the difference between Relatives and Family?

Table 6: The associations have to be exposed through Odds Ratios, it is not understood if the differences are between types of vaccines or between categories.
Table 7 is very interesting but in logistic regression it is not correct to introduce all the variables at once. 

Once the different errors have been corrected, I will be able to review the discussion and conclusions section.

Author Response

Reviewer 1

Review the entire document: vaccine hesitancy has defined vaccine hesitancy as a delay in acceptance or refusal of vaccines despite availability of vaccination services.  

Reply: As pointed out by the respected reviewer, Yes we used the WHO definition. The World Health Organization (WHO)’s Strategic Advisory Group of Experts on Immunization defines vaccine hesitancy as a “delay in acceptance or refusal of vaccines despite availability of vaccination services” Reference : World Health Organization. Report of the SAGE Working Group on Vaccine Hesitancy.  https://www.who.int/ immunization/sage/meetings/2014/october/1_Report_WORKING_GROUP_vaccine_hesitancy_final.pdf (accessed on 20 May 2021).

Other reference: Available online: https://www.who.int/news/item/18-08-2015-vaccine-hesitancy-a-growing-challenge-for-immunization-programmes

Abstract:

In the results section: the data shown are confusing, I understand that 161 of 206 women interviewed were vaccinated, but 12 were hesitant with being vaccinated, as expressed corresponds to 6% and not 26.6% (12/45). The comparison between pregnant and lactating women is better through the odds ratio than through simple percentages.

There are contradictory data, I think due to the definition of "vaccine hesitancy". The WHO considers hesitancy if the time of vaccination has been delayed or not vaccinated. Therefore the vaccine hesitancy is (12+176)/206 = 91.3%.

In view of the results, it seems that strategies should be improved to avoid the delay in vaccination, although compulsory vaccination has been a very important factor.

Reply:  As pointed out by the respected reviewer, A total of that 161 of 206 (78.2%) women were vaccinated and 45 (21.8%) women did not receive the vaccine. The WHO considers hesitancy if the time of vaccination has been delayed or not vaccinated. Overall, 36 (17.4%) unvaccinated women were currently hesitant to receive the vaccine. A higher proportion of the subjects (35.5%, 16/45) were willing to get vaccinated after delivery and 5/45 (11.1%) were willing to get vaccinated after breast feeding .  Additionally, a moderate proportion (3/45, 6.6%) were willing to get vaccinated before traveling, and nearly 26.6% (12/45) had decided not to receive the COVID-19 vaccine. Thus 12 refused+ 24 delayed =36

Overall 192 out of 206 (93.2%) females had delayed their vaccination due to uncertainty of the safety of the vaccine and some other medical reason at some point of time (Table 5).  At the time of survey already 161 respondents were vaccinated.  The most reason perceived for delaying COVID-19 vaccination was being worried about the side effects (176, 85.44%) and the effects on the fetus (130, 63.1%) in the sample population (n=206). 

Objective:

It looks like KAP (knowledge, attitudes and practices) are the objectives of this study, however knowledge has not been collected in this study, only practices (Vaccinate, delayed vaccinate, don't vaccinate) and attitudes are reflected. The other objective of the study is concerns, although this is actually an attitude. What has been collected and is not mentioned in this objective section are the sources of information (Mass Media/ Friends/ Family/ Wealth care workers).

Reply: As pointed by the respected reviewer we have removed the words knowledge and concerns from our study objective and revised it.

Methodology

The study design is not qualitative; qualitative studies have another approach where participants are interviewed in a relaxed manner using different techniques.  This study is simply a cross-sectional study.  Please add the Odds Ratio of the comparisons made.

Reply: As suggested by the respected reviewer we have removed the word qualitative.

Please mention the type of regression performed which is subsequently shown in Table 7.

Reply: Multinomial logistic regression to estimate the parameters was applied. This has been mentioned as suggested.

Results:

Regarding table 1: Does "Master" mean graduate? And does "Government" mean civil servant?

Reply: Masters means postgraduate and yes government means civil servant

Regarding table 3: The row "Abortion or still birth" it is not clear the statistical test(s) performed to obtain statistical significance.

Reply: This has been revised as pointed.

Considering that, the SAGE Working Group has defined vaccine hesitancy as a delay in acceptance or refusal of vaccines despite availability of vaccination services (https://www.euro.who.int/__data/assets/pdf_file/0004/329647/Vaccines-and-trust.PDF), the paper should be reviewed and it should be noted that the results do not support the so-called "protection motivation theory" in boosting vaccine acceptance.

Table 5: There is a distinction between family and husband, so what is the difference between Relatives and Family?

Reply: We have revised the terms as close family members (spouse, child, sibling and parent) and distant relatives.

Table 6: The associations have to be exposed through Odds Ratios, it is not understood if the differences are between types of vaccines or between categories.

Reply: As suggested this has been revised.

Table 7 is very interesting but in logistic regression it is not correct to introduce all the variables at once.

Reply: No changes done.

Reviewer 2 Report

The vaccination against COVID-19 among Saudi women is similar to the global pattern; It is uncovered high levels of hesitancy are about one-fourth. The hesitation is primarily induced due to adverse effects of the vaccine and social media-related misinformation. The participants' primary facilitator of vaccine acceptance was a belief in protection from COVID-19 but concern about possible side effects on themselves or their baby's health.

Title: The title is easy to follow and has no mistakes.

Abstract: This section was well-written and easy to understand.

Introduction: Seem to be fine
Results: All the part of this section is easy to follow, and no mistakes.
Discussion: The discussion of the study is very comprehensive compared with other similar studies. Describe the probable mechanism of other vaccines given in pregnancy. Describe the list of COVID-19 vaccines claimed safe for Pregnant and breastfeeding women
Conclusion: In this part of the manuscript, the authors have successfully established the conclusion of their findings.

References: The core references are acceptable but significantly lesser as per the journal's standard.

Author Response

The vaccination against COVID-19 among Saudi women is similar to the global pattern; It is uncovered high levels of hesitancy are about one-fourth. The hesitation is primarily induced due to adverse effects of the vaccine and social media-related misinformation. The participants' primary facilitator of vaccine acceptance was a belief in protection from COVID-19 but concern about possible side effects on themselves or their baby's health.

Title: The title is easy to follow and has no mistakes.

Abstract: This section was well-written and easy to understand.

Introduction: Seem to be fine
Results: All the part of this section is easy to follow, and no mistakes.
Discussion: The discussion of the study is very comprehensive compared with other similar studies. Describe the probable mechanism of other vaccines given in pregnancy. Describe the list of COVID-19 vaccines claimed safe for Pregnant and breastfeeding women

Reply: As suggested by the respected reviewer we have included some more references please see references 7.8, 9 37, 38, 39 and 40.

As to the best of our knowledge all available vaccines by that time were considered safe to pregnant women. We have mentioned types of vaccines that were available in KSA and we added one option for others if some women might take it from outside.

Conclusion: In this part of the manuscript, the authors have successfully established the conclusion of their findings.

References: The core references are acceptable but significantly lesser as per the journal's standard.

Reply: As suggested the total count is 40

Reviewer 3 Report

I would request the authors to use the term pregnant women than perinatal women.

Page 2 line 51, it is mentioned that many reports but you used only one study but not many studies so you need to change this

The background should clearly highlight the vaccine hesitancy among the general population and some recent evidence of vaccine hesitancy from Saudi and the middle east region should be addressed. Then the author should discuss the issues with pregnant women.

How the practice can be assessed for vaccine hesitancy? This is completely wrong. This is not something regular taking place. Hence why practice was part of the research aim.

The abstract says an observational study which is an epidemiological study design but the method says it is a qualitative cross-sectional study.

More detailed methodological information is needed. The sample selection process is not clear and how the questionnaire was validated? Was there any pilot conducted? What are the key measures used? The data analysis plan should be explained clearly.

Author Response

I would request the authors to use the term pregnant women than perinatal women.

Reply: As suggested by the respected reviewer the term pregnant women has been used.

Page 2 line 51, it is mentioned that many reports but you used only one study but not many studies so you need to change this

Reply: As pointed we have removed the word researchers

The background should clearly highlight the vaccine hesitancy among the general population and some recent evidence of vaccine hesitancy from Saudi and the middle east region should be addressed. Then the author should discuss the issues with pregnant women.

Reply: As pointed we have revised  it. Please see references 7.8 and 9

How the practice can be assessed for vaccine hesitancy? This is completely wrong. This is not something regular taking place. Hence why practice was part of the research aim.

Reply:  The main objective of the current study was to assess levels of attitudes and practices of women in the pregnancy period regarding COVID-19 vaccination.

The abstract says an observational study which is an epidemiological study design but the method says it is a qualitative cross-sectional study.

Reply: As suggested by the respected reviewer we have removed the word qualitative.

More detailed methodological information is needed. The sample selection process is not clear and how the questionnaire was validated? Was there any pilot conducted? What are the key measures used? The data analysis plan should be explained clearly.

A pilot survey was conducted on 30 pregnant females before initiating the actual data collection; however, these pilot samples were excluded from the final sample size. The prime objective of the pilot survey was to guarantee the validity and reliability of the questionnaire. The face and content validity of the questionnaire was assessed by specialists in the field of research. Face validity was evaluated through the review and comments offered by a panel of experts related to readability, clarity of wording, layout, and feasibility of the questionnaire. Content validity was evaluated by the content validity index, which is the mean content validity ratio of all questions in a questionnaire.

Round 2

Reviewer 1 Report

no comments

Author Response

Thank you

Reviewer 2 Report

Seem to be fit for publication

Author Response

Thank you

Reviewer 3 Report

Good to see you have addressed most of the queries, but still I don't see the use of so many recent articles from SA included here and then you need to show the research gap properly. 

Author Response

As suggested by the respected reviewer, we have mentioned some recent articles from SA. There are not many recent articles from SA on pregnant females. Please see references 8, 11,12, 16, 17, 23 and 37.  Also the research gap has been mentioned (lines 305-307).

Thank you